# HG-Mamba: Heuristic-Guided State Space Model for Laparoscopic Image Desmoking

## Abstract

Developing smoke removal algorithms for laparoscopic surgery is crucial for enhancing surgical visibility and supporting accurate intraoperative decision-making. Recently, Mamba, a representative state space model (SSM), has shown strong potential in visual tasks by balancing global receptive fields with efficiency. However, its reliance on sequential state transitions limits spatial correlation modeling, and its feed-forward layers lack mechanisms to model frequency features—both of which hinder effective removal of complex surgical smoke. To overcome these limitations, we propose Heuristic-Guided Mamba (HG-Mamba), which extends Mamba by integrating spatial and frequency domain improvements. HG-Mamba comprises two key components: a Heuristic-Guided State Space Model (HG-SSM), which performs input-guided dynamic sampling and flexible state fusion to enable adaptive spatial context modeling; and a Frequency Refine Feed-Forward Network (FR-FFN), which conducts multi-band frequency decomposition and adaptive weighting to enhance frequency-domain representations. By jointly leveraging spatial adaptability and frequency-aware refinement, HG-Mamba serves as an effective backbone for surgical smoke removal. Extensive experiments demonstrate that HG-Mamba outperforms state-of-the-art methods on both synthetic and real-world smoke/smokeless datasets. The code will be publicly released.

## 1 Introduction

Laparoscopic surgery is a widely used minimally invasive technique that offers advantages such as smaller incisions, reduced postoperative pain, and lower infection risk (Sauerland et al., 2010). However, energy instruments like electrocautery and lasers generate substantial surgical smoke, significantly degrading the clarity of the operative field. This visual obstruction hampers the surgeon's ability to perceive anatomical structures, compromising intraoperative judgment and reducing surgical safety and success rates (Azam et al., 2022). Since rapid smoke evacuation during surgery is often difficult to achieve, post-processing laparoscopic images to eliminate surgical smoke helps enhance intraoperative visual clarity in a convenient and efficient manner (Chen et al., 2020).

The dynamic and spatially complex nature of surgical smoke challenges existing desmoking methods (Chen et al., 2020; Zhou et al., 2022; Hong et al., 2023), motivating the need for more effective global context modeling. Recently, state space models(SSMs) (Gu et al., 2022b; Smith et al., 2022; Mehta et al., 2022) have gained increasing attention as efficient alternatives to traditional architectures such as Transformers (Vaswani et al., 2017), due to their ability to model long-range dependencies with linear complexity. Among them, Mamba (Gu & Dao, 2023), a structured SSM, introduces an improved selection mechanism that enables the retention of relevant information while suppressing irrelevant content. Benefiting from these capabilities, Mamba has achieved remarkable performance in various vision tasks, including image restoration (Guo et al., 2025b; Zou et al., 2024; Guo et al., 2025a) and segmentation (Liang et al., 2025; Wang et al., 2025), suggesting its potential for surgical smoke removal, which requires both global context modeling and computational efficiency.

However, to the best of our knowledge, the application of Mamba to laparoscopic surgical smoke removal has not yet been explored. Directly applying Mamba to this task poses two key challenges: (1) its state updates operate along one-dimensional sequences, limiting its capacity to model spatial

Figure 1: (a) The original unidirectional scan strategy and its modified versions remain limited to sequential state dependencies. (b) In contrast,our HG-SSM introduces dynamic dependencies, where an input-guided dynamic sampling strategy selects spatially related states for fusion.

correlations inherent in two-dimensional surgical images; and (2) its feed-forward network, based on MLPs, overlooks frequency-domain characteristics, which are often relevant for modeling fine structures in degraded images (Zou et al., 2024; Kong et al., 2025).

The first challenge has been partly tackled in general image processing tasks by converting 2D images into 1D sequences using fixed scanning strategies such as sweep scan (Liu et al., 2024), local scan (Huang et al., 2025), and continuous scan (Yang et al., 2024). However, these predefined patterns struggle to capture the complex spatial structures in real-world images, and multi-directional scans often increase computational cost. More recent efforts have improved adaptability—DefMamba(Liu et al., 2025) introduces learnable scanning paths, while MambaIRv2 (Guo et al., 2025a) proposes a semantic-aware unfolding strategy. Nonetheless, all these methods still rely on sequential state transitions (as shown in Fig. 1a), which inherently limit the flow of contextual information across spatial dimensions. Unlike these methods, Spatial-Mamba (Xiao et al., 2025) introduces a structure-aware state fusion mechanism that aggregates neighboring state variables via fixed convolutions in the latent state space to enhance spatial context modeling. However, such fixed fusion strategies may struggle to accommodate the dynamic and irregular nature of surgical smoke, potentially leading to suboptimal performance in complex intraoperative scenes.

In response, we propose a novel Heuristic-Guided State Space Model (HG-SSM) that enables flexible state variable fusion through a dynamic sampling strategy (as shown in Fig. 1b). While the state variable matrix in conventional SSMs tends to disrupt the spatial structure of the input, the input itself preserves rich spatial information. Leveraging this, we leverage the spatial semantics preserved in the input to heuristically predict spatially related contextual states and establish dynamic dependencies by fusing them. This design enhances the model's capability to flexibly capture spatial context information.

To address the second challenge, we introduce the Frequency Refine Feed-Forward Network (FR-FFN). Unlike coarse frequency selection methods (Kong et al., 2023; 2025), FR-FFN decomposes features into multiple frequency bands and independently modulates each band with learned adaptive weights, facilitating enhanced restoration of images affected by surgical smoke.

FR-FFN complements HG-SSM, and together they synergistically improve smoke removal performance by jointly exploiting spatial adaptability and frequency-aware refinement. By integrating these two modules, we develop Heuristic-Guided Mamba (HG-Mamba), serving as the backbone of our smoke removal network to enable accurate and effective desmoking.

In addition, although the first in vivo paired dataset (Xia et al., 2025) has recently been released, large-scale and diverse paired smoke/smokeless data remain scarce for laparoscopic image desmoking. This scarcity mainly arises from the difficulty of collecting in vivo paired images due to the complexity of surgical environments. To complement the limited real data, we construct a synthetic paired dataset by rendering smoke with Blender and compositing it onto real laparoscopic images. The resulting dataset provides diverse surgical scenes and complex smoke patterns, enabling more comprehensive training and evaluation of desmoking methods. Our contributions are summarized as follows:

- HG-SSM is proposed, utilizing dynamic sampling and state fusion to overcome the sequential transition bottleneck for flexible context modeling.

- FR-FFN is designed to enhance feature representation through multi-frequency decomposition and dynamic weighting, improving desmoking performance.

- HG-Mamba, a novel laparoscopic desmoking backbone combining HG-SSM and FR-FFN, achieves superior performance with only 1.69M parameters, reducing the parameter count by 82.45% compared to MambaIRv2-S (9.63M).

- A large synthetic smoke dataset is constructed, and extensive experiments demonstrate HG-Mamba's superior performance over state-of-the-art methods on both synthetic and real surgical smoke data.

## 2 RELATED WORKS

### 2.1 LAPAROSCOPIC IMAGE DESMOKING

In recent years, various methods have been proposed to tackle the challenge of smoke removal in laparoscopic images. Traditional approaches (Tchaka et al., 2017; Wang et al., 2018) are mostly based on the atmospheric scattering model, which restores clear images by estimating handcrafted priors. However, these methods often suffer from color distortions and structural artifacts. With the advancement of deep learning, researchers have increasingly adopted end-to-end trainable neural networks that learn the desmoking process directly from data (Chen et al., 2020; Pan et al., 2022; Zhou et al., 2022; Li et al., 2024), thereby reducing reliance on prior assumptions. For instance, Chen et al. (2020) modifies the U-Net architecture to construct a two-stage desmoking framework, integrating a smoke detection module that assists in accurately identifying and restoring smoke-occluded regions. To address the lack of real paired data, Pan et al. (2022) and Zhou et al. (2022) utilize CycleGAN-based unpaired training strategies to translate images between smoke and smoke-free domains. Wang et al. (2023) introduce the Swin Transformer to enhance feature representation. Li et al. (2024) design a desmoking network based on the diffusion model to enhance the restoration of details in smoke-affected regions.

These methods are primarily built on CNN or Transformer backbones. While CNNs are computationally efficient, their limited receptive field restricts their ability to capture global smoke distributions. Transformers offer strong global modeling capabilities but incur quadratic computational complexity with the number of tokens, making them less suitable for time-sensitive and resource-constrained surgical environments. Given the limited memory of laparoscopic devices and the urgency of intraoperative decision-making, developing lightweight and effective desmoking methods remains a critical objective.

### 2.2 STATE SPACE MODELS

SSMs (Gu et al., 2022b; Smith et al., 2022; Mehta et al., 2022) are fully recurrent architectures designed for sequence modeling, which have recently achieved notable progress in both structural design and representational capacity. Several enhanced variants of SSMs (Gu et al., 2022a;b; Gu & Dao, 2023) have reached performance comparable to that of Transformers, while maintaining linear computational complexity. Among these developments, Mamba (Gu & Dao, 2023) has emerged as a pivotal milestone. With its distinctive selective mechanism and hardware-efficient implementation, Mamba has demonstrated performance that rivals or even surpasses state-of-the-art Transformer models on various one-dimensional sequence modeling tasks.

Building on its success in sequential domains, Mamba has been adapted to computer vision tasks. To handle the two-dimensional nature of images, Vim (Zhu et al., 2024) and VMamba (Liu et al., 2024) extended Mamba's 1D scanning mechanism to bidirectional and four-directional cross-scanning, respectively. DefMamba (Liu et al., 2025) introduces a deformable scanning mechanism that dynamically adjusts the scanning path to better capture spatial variations in input features. MambaIRv2 (Guo et al., 2025a) proposes a semantic-guided unfolding strategy to group semantically similar pixels into 1D sequences, thereby enhancing semantic-level modeling. Even so, these modified scanning strategies do not alter the sequential dependencies between hidden states.

Unlike these scan-based adaptations, Spatial-Mamba (Xiao et al., 2025) follows a different paradigm by applying unidirectional scanning to convert visual inputs into sequences, computing latent state

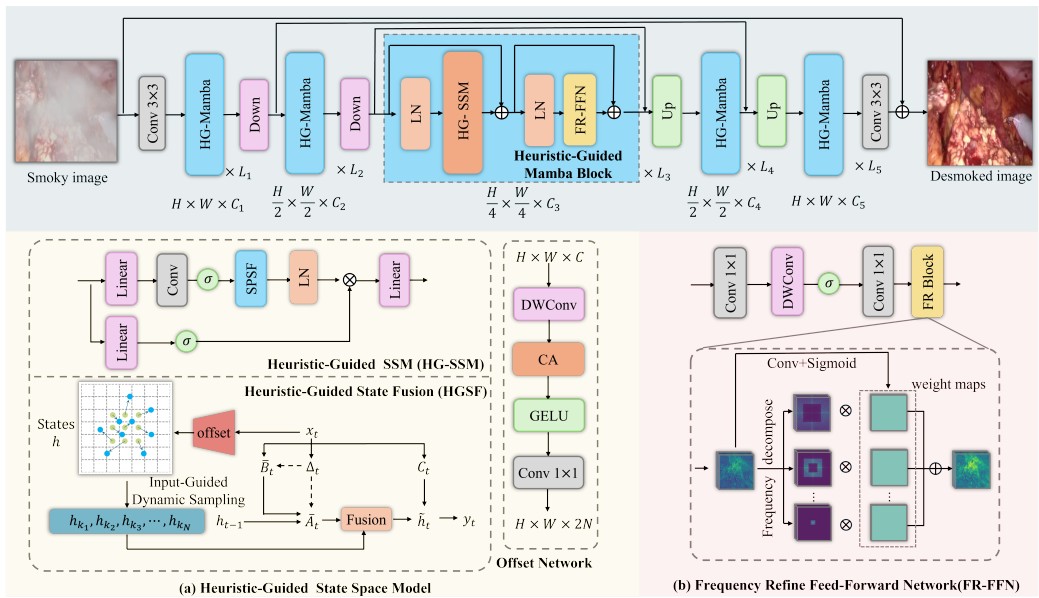

Figure 2: The overall architecture of our HG-Mamba-based desmoking network.

variables, and reshaping them into 2D representations. It then fuses neighboring state variables using fixed dilated convolutions, enabling structure-aware state fusion and reducing the reliance on sequential dependencies. However, the fixed convolution kernel implies a static sampling strategy, which limits its adaptability to the highly dynamic and non-uniform smoke patterns commonly observed in laparoscopic surgery, and may lead to suboptimal desmoking performance. In contrast, our HG-SSM heuristically selects spatially related states based on the spatial semantics of the input and fuses them to improve dependencies between states, which enables it to more effectively attend to smoke-affected regions.

## 3 METHODS

### 3.1 SSM PRELIMINARIES

SSMs like Mamba (Gu & Dao, 2023) are discretized for efficient training. In Mamba, the discrete state transition and observation equations of the SSM are:

$$h_t = \overline{\boldsymbol{A}}_t h_{t-1} + \overline{\boldsymbol{B}}_t x_t, y_t = \boldsymbol{C}_t h_t, \tag{1}$$

where $h_{t-1}$ and $h_t \in \mathbb{R}^N$ denote the hidden state at the previous and current time steps, respectively. $x_t \in \mathbb{R}^L$ and $y_t \in \mathbb{R}^L$ are the input/output at time $t$. $\overline{\boldsymbol{A}}_t = \exp(\Delta_t \boldsymbol{A})$ is the control matrix. $\overline{\boldsymbol{B}}_t = (\Delta_t \boldsymbol{A})^{-1}(\exp(\Delta_t \boldsymbol{A}) - \boldsymbol{I}) \cdot \Delta_t \boldsymbol{B}_t$ and $\boldsymbol{C}_t$ are input and output projection matrices.

### 3.2 HEURISTIC-GUIDED STATE SPACE MODEL

At time step $t$, we assume $\overline{\boldsymbol{A}}_{t+1} = \boldsymbol{I}$. Based on Eq. 1, the recursive form of $h_t$ can be derived as follows:

$$\begin{aligned} h_t &= \overline{\boldsymbol{A}}_t(\overline{\boldsymbol{A}}_{t-1} h_{t-2} + \overline{\boldsymbol{B}}_{t-1} x_{t-1}) + \overline{\boldsymbol{B}}_t x_t \\ &= (\textstyle\prod_{i=t-1}^{t} \overline{\boldsymbol{A}}_i) h_{t-2} + \textstyle\sum_{i=t-1}^{t} \overline{\boldsymbol{A}}_{i+1} \overline{\boldsymbol{B}}_i x_i \\ &= ... \\ &= (\textstyle\prod_{i=1}^{t} \overline{\boldsymbol{A}}_i) h_0 + \textstyle\sum_{i=1}^{t} \overline{\boldsymbol{A}}_{i+1} \overline{\boldsymbol{B}}_i x_i. \end{aligned} \tag{2}$$

It can be observed that the state $h_t$ depends on $h_t - 1$ and implicitly depends on the historical states $h_{t-2}, ..., h_1, h_0$, but is independent of future states $h_k(k > t)$. This strict sequential dependency enforces that SSM computations rely on ordered state transitions, which limits the information flow

between state variables. Moreover, $h_t$ only depends on historical input tokens $x_i(i \leq t)$, but for originally 2D inputs, this results in a loss of spatial structure. Simply altering the scanning strategy only reorders the mapping of input tokens, i.e., $x_i \rightarrow x_j$. This merely substitutes $x_i$ with $x_j$ in the Eq. 2, leaving the sequential state transitions unchanged.

To build dynamic dependencies between states, we propose the HG-SSM, as illustrated in Fig. 2a. Its core is the Heuristic-Guided State Fusion (HGSF) module, which introduces input-guided dynamic dependencies among state variables. The process is described in three steps: 1) **Sequence construction**: a unidirectional scanning strategy is used to flatten the 2D image into a 1D sequence $x_t$. The corresponding hidden state $h_t$ is then computed using the equation $h_t = \overline{\boldsymbol{A}}_t h_{t-1} + \overline{\boldsymbol{B}}_t x_t$. 2) **Spatial reshaping**: all the hidden states $h_t$ are reshaped into 2D spatial map $h$ aligned with the input resolution to enable spatial sampling. 3) **State sampling and fusion**: the sampled hidden states for $h_t$ are obtained via the function $\mathcal{S}(h_t|x)$ (see Alg. 1), which selects spatially relevant states for each position in $h_t$ based on the input $x$. The selected states are then aggregated via a weighted summation with weights $\omega_{k_i}$, producing the updated hidden state $\tilde{h}_t = \sum_{k_i \in \mathcal{S}(h_t|x)}^n \omega_{k_i} h_{k_i}$.

---

**Algorithm 1** Input-Guided Dynamic Sampling for $h_t$

---

**Require:** Hidden state map $h \in \mathbb{R}^{H \times W \times C}$, input feature map $x \in \mathbb{R}^{H \times W \times C}$, number of sampling points $N$
**Ensure:** Sampling set $\mathcal{S}(h_t|x) = \{(k_i, h_{k_i})\}$ for $h_t$
 1: Define a fixed local sampling grid $\{p_i^0\}_{i=1}^N$ around the spatial location of $h_t$ in $h$
 2: Predict input-dependent offsets $\{\Delta p_i[x]\}_{i=1}^N$ from input $x$ using an offset network
 3: **for** $i = 1$ to $N$ **do**
 4:     Compute the sampling location $p_i = p_i^0 + \Delta p_i[x]$
 5:     Obtain the sampled hidden state $h_{k_i}$ via bilinear interpolation $\Psi(h, p_i)$.
 6:     Store $(k_i, h_{k_i})$ in $\mathcal{S}(h_t|x)$
 7: **end for**
 8: **return** $\mathcal{S}(h_t|x)$

---

With this sampling mechanism, the state transition equation of HG-SSM can be rewritten as:

$$\tilde{h}_t = \sum_{k_i \in \mathcal{S}(h_t|x)}^n \omega_{k_i} \overline{\boldsymbol{A}}_{k_i-1} h_{k_i-1} + \sum_{k_i \in \mathcal{S}(h_t|x)}^n \omega_{k_i} \overline{\boldsymbol{B}}_{k_i} x_{k_i}. \tag{3}$$

Since $k_i - 1 > t$ is allowed, the state $\tilde{h}_t$ is obtained by aggregating hidden states $h_{k_i-1}$ and input tokens $x_{k_i}$ from arbitrary time steps, establishing dynamic dependencies beyond the historical sequence. Using $x$ as guidance is crucial because recursive computation of hidden states tends to disrupt the spatial structure of the original input, whereas $x$ retains rich spatial semantics. By guiding the state fusion with $x$, the model facilitates dynamic spatial interactions among state variables, thereby enhancing contextual information flow and improving focus on smoke-affected regions. To predict the input-dependent sampling offsets, a lightweight offset network is used (see Fig. 2a): it applies a depthwise convolution to extract local spatial features, a Channel Attention (CA) mechanism (Hu et al., 2018) to integrate global contextual information, followed by a $GELU$ activation and a $1 \times 1$ convolution to produce the offsets.

The output response is produced through the observation equation:

$$y_t = C_t \tilde{h}_t, \tag{4}$$

where the output incorporates richer spatial features. The HGSF module, together with the observation equation in Eq. 4, defines the complete HG-SSM. By leveraging heuristic-guided sampling and fusion, HG-SSM establishes flexible state interactions and preserves spatial semantics, thereby facilitating modeling of spatially-varying smoke patterns in laparoscopic images.

### 3.3 FREQUENCY REFINE FEED-FORWARD NETWORK

While prior works, such as DFFN (Kong et al., 2023) and EDFFN (Kong et al., 2025), have introduced frequency-domain operations into FFNs for image restoration, they primarily focus on coarse frequency selection, which limits the effective utilization of frequency information and potentially

affects reconstruction quality and structural preservation. To address this limitation, we propose the Frequency Refine Feed-Forward Network (FR-FFN), which modulates the frequency content in a finer-grained manner, as shown in Fig. 2b. This module first utilizes a lightweight feed-forward sub-network to extract refined spatial representations from the input features. The extracted features are then transformed into the frequency domain using the Discrete Fourier Transform (DFT) and decomposed into multiple frequency bands using binary masks defined by predefined thresholds. For each frequency band, a learnable weight map is predicted to dynamically modulate its frequency components, enabling expressive frequency reconstruction and improved structural preservation. The FR-FFN can be described by the following formulations:

$$\tilde{X} = Conv_{1 \times 1}(GELU(DWConv(Conv_{1 \times 1}(X)))), \tag{5}$$

$$Y_b = Y_{b-1} - \mathcal{F}^{-1}(M_b \odot \mathcal{F}(\tilde{X})), Y_0 = \tilde{X}, \tag{6}$$

$$M_b^{i,j} = \begin{cases} 1, & \text{if } \psi_b \leqslant \max(|u|, |v|) < \psi_{b+1} \\ 0, & \text{otherwise} \end{cases}, \tag{7}$$

$$\tilde{Y} = \sum_{b=0}^{B-1} W_b \odot Y_b, \tag{8}$$

where $X$ is the input feature, and $\mathcal{F}$ denotes the DFT. $M_b$ corresponding to the mask of the $b$-th frequency band, defined by the set of frequency thresholds $\{0, \psi_1, \psi_2, ..., \psi_{B-1}, \frac{1}{2}\}$. $Y_b$ represents the frequency component in the $b$-th band, and $W_b$ is the learnable weight map derived from $\tilde{X}$ that reweights $Y_b$ for adaptive enhancement. This design enables adaptive frequency modulation to better preserve structures and restore details in smoke-affected regions. Similar to (Chen et al., 2024a), we adopt the octave-wise division strategy to decompose the frequency spectrum into multiple bands, with thresholds set as $\{0, \frac{1}{16}, \frac{1}{8}, \frac{1}{4}, \frac{1}{2}\}$.

### 3.4 Heuristic-Guided Mamba and Training Loss

By integrating the HG-SSM and FR-FFN modules, we construct HG-Mamba as the backbone of our desmoking network, as illustrated in Fig. 2. HG-Mamba effectively combines spatial and frequency-domain characteristics to achieve accurate smoke removal. The network is composed of five stages, with channel dimensions set to $[C_1, C_2, C_3, C_4, C_5] = [48, 64, 128, 64, 48]$ and the number of HG-Mamba blocks in each stage configured as $[L_1, L_2, L_3, L_4, L_5] = [2, 3, 4, 3, 2]$. This configuration follows a symmetric encoder-decoder pattern and is designed to balance representation capacity with parameter efficiency.

To train our desmoking framework, we employ a combination of the pixel-wise L1 loss and the Frequency-domain Contrastive Regularization (FCR) loss (Gao et al., 2024). The overall training objective is formulated as:

$$\mathcal{L} = \mathcal{L}_1 + \alpha \cdot \mathcal{L}_{FCR}, \tag{9}$$

where $\alpha$ is a balancing coefficient, empirically set to 0.1 in our experiments.

## 4 Experiment

### 4.1 Datasets

DesmokeData (Xia et al., 2025) is the first publicly released in vivo paired dataset for laparoscopic smoke removal, comprising 3,464 smoke/smoke-free image pairs from 21 surgical scenes. We split the dataset into training, validation, and test sets at a $70\%/15\%/15\%$ ratio.

Due to the difficulty of collecting in vivo paired images, we construct a large-scale synthetic dataset by simulating laparoscopic surgical smoke to provide additional data for training and for evaluating models under varied smoke patterns. Specifically, we sample 15,000 smoke-free images from 91 videos in the Cholec80 dataset (Twinanda et al., 2017) and the Hamlyn Centre Laparoscopic/Endoscopic Video (Ye et al., 2017) datasets. We then generate 10,000 smoke masks using the Blender rendering engine. By adjusting key rendering parameters (e.g., density, vorticity, and heat), we produce smoke masks with diverse appearances to increase variability and realism. These masks are subsequently overlaid onto smoke-free images using alpha blending to generate the corresponding smoky images. Among them, 5,000 unique masks were used to synthesize 10,000 training

Table 1: Quantitative comparison of various methods on the synthetic dataset and the DesmokeData benchmark. ↓ denotes that lower values are better, while ↑ indicates that higher values are better. Underlined values highlight second-best results, and bold values indicate the best performance.

| Methods | Venue | Synthetic Dataset | | | DesmokeData | | | Overhead | |
|---|---|---|---|---|---|---|---|---|---|
| | | SSIM↑ | PSNR↑ | CIEDE↓ | SSIM↑ | PSNR↑ | CIEDE↓ | Params↓ | MACs↓ |
| DCP | TPAMI'11 | 0.841 | 18.886 | 9.963 | 0.763 | 19.444 | 9.872 | - | - |
| Cyclic-DeGAN | CBM'20 | 0.865 | 21.280 | 7.525 | 0.833 | 22.397 | 7.173 | 11.97M | 28.11G |
| FFA-Net | AAAI'20 | 0.956 | 28.778 | 2.932 | 0.874 | 26.127 | 4.563 | 4.46M | 288.86G |
| AECR-Net | CVPR'21 | 0.949 | 27.487 | 3.696 | 0.870 | 25.751 | 4.859 | 2.61M | 42.93G |
| DS-CycleGAN | TCBB'22 | 0.792 | 20.412 | 9.495 | 0.766 | 21.927 | 7.688 | 30.81M | 136.16G |
| DehazeFormer-B | TIP'23 | 0.959 | 28.219 | 3.159 | 0.879 | 26.341 | 4.511 | 2.52M | 19.76G |
| DEA-Net | TIP'23 | 0.964 | 29.706 | 2.833 | 0.847 | 25.747 | 4.714 | 3.65M | 24.68G |
| ConvIR-S | TPAMI'24 | 0.953 | 28.849 | 2.891 | 0.877 | 26.442 | 4.540 | 5.53M | 42.23G |
| MB-Taylor-B V2 | TPAMI'25 | 0.969 | 29.551 | 2.658 | 0.878 | 26.515 | 4.383 | 2.63M | 24.40G |
| SGDN | AAAI'25 | 0.970 | 29.670 | 2.605 | 0.879 | 26.871 | 4.168 | 11.09M | 41.16G |
| MambaIRv2-S | CVPR'25 | 0.964 | 28.730 | 2.814 | 0.870 | 25.234 | 4.891 | 9.63M | 192.91G |
| Ours | - | 0.972 | 30.393 | 2.354 | 0.888 | 27.072 | 4.064 | 1.69M | 18.62G |

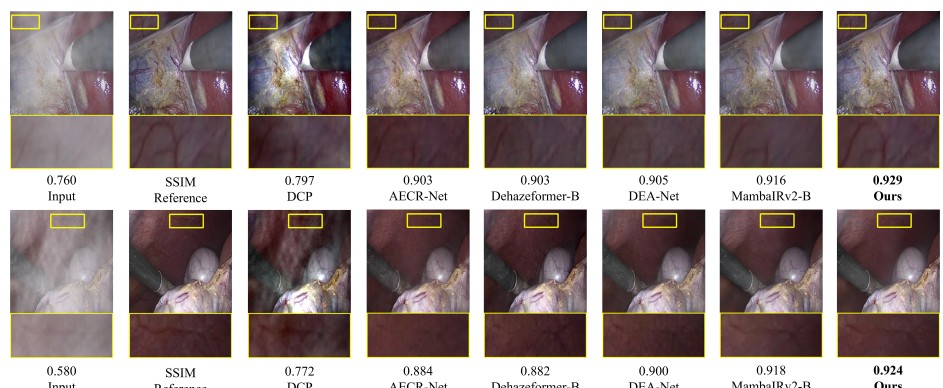

Figure 3: Visual comparison results on the synthetic dataset.

images, while the remaining 5,000 were split into validation and test sets in a 3:2 ratio. All sets were constructed from different video clips to prevent data leakage. As a result, the final training, validation, and test sets consist of 10,000, 3,000, and 2,000 paired images, respectively.

To verify the distributional characteristics of synthetic data, we visualize the real and synthetic samples using t-SNE in the feature space, as shown in Fig. 5a. While some discrepancies between the distributions are observed, a notable degree of overlap is evident. This overlap indicates that the synthetic dataset reasonably approximates the key properties of real surgical smoke, making it a useful data source for model training and evaluation.

## 4.2 EXPERIMENTAL SETTINGS

We compare our method against 11 representative approaches, including: one traditional method: DCP (He et al., 2010), two desmoking methods: Cyclic-DeGAN (Venkatesh et al., 2020) and DS-CycleGAN (Zhou et al., 2022), five dehazing models: FFA-Net (Qin et al., 2020), AECR-Net (Wu et al., 2021), DehazeFormer-B (Song et al., 2023), DEA-Net (Chen et al., 2024b), and SGDN (Fang et al., 2025), and three general image restoration models: ConvIR-S (Cui et al., 2024), MB-Taylor-B V2 (Jin et al., 2025), and MambaIRv2-S (Guo et al., 2025a). The evaluation metrics include commonly used full-reference image quality measures: SSIM, PSNR, and CIEDE-2000 (referred to as CIEDE for simplicity).

To ensure a fair comparison, all models are retrained on both the synthetic dataset and Desmoke-Data using their original parameter configurations. On the synthetic dataset, all input images are resized to $224 \times 224$ to facilitate efficient training, validation, and testing. All models are trained for 100 epochs. On DesmokeData, inputs are randomly cropped to $192 \times 192$, with additional data augmentation techniques including random rotation and flipping. During validation and testing, the

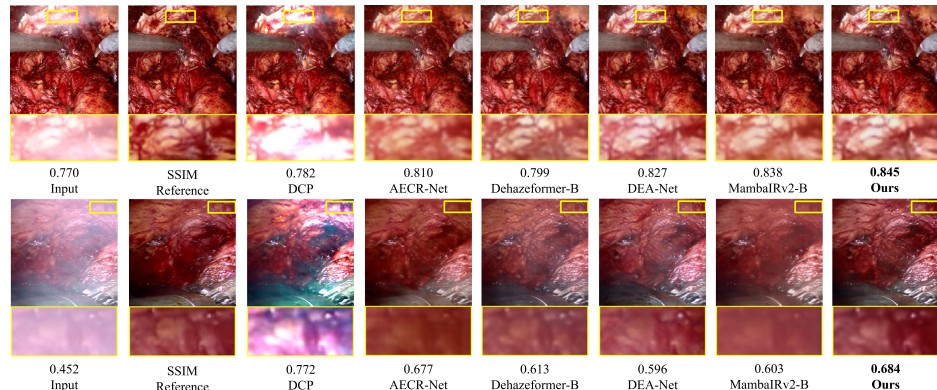

Figure 4: Visual comparison results on the DesmokeData.

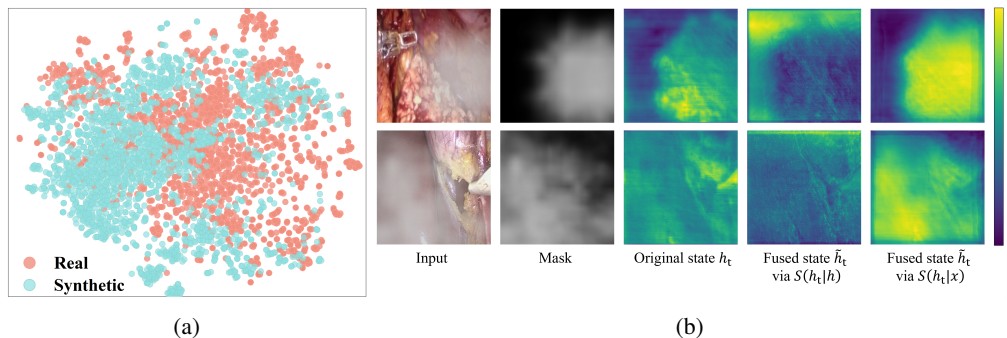

Figure 5: (a) t-SNE visualization of synthetic and real smoke images based on Inception-V3 (Szegedy et al., 2016) deep features. (b) Visualization of hidden state before and after fusion (channel-averaged and normalized to [0,1]).

input size is fixed at $192 \times 192$. All models are trained for 500 epochs. The best-performing model checkpoint is selected based on SSIM performance on the validation set.

Our model is optimized using the AdamW optimizer, and batch size is set to 4. On the synthetic dataset, the learning rate decays from $4e{-4}$ to $4e{-6}$ following the cosine annealing schedule (He et al., 2019), while a fixed learning rate of $1e{-4}$ is used on DesmokeData. All experiments are conducted on a single NVIDIA A100 GPU.

### 4.3 COMPARISON ON SYNTHETIC DATASET

As shown in Table. 1, our method outperforms all state-of-the-art (SOTA) approaches across all metrics. Specifically, compared to MambaIRv2-S, our model achieves improvements of +0.83% in SSIM, +5.79% in PSNR, and +16.35% in CIEDE, while reducing the parameter count by 82.45%. In addition, Fig.3 presents the visual desmoking results of compared methods. DCP suffers from inaccurate priors, leading to noticeable color distortion and structural artifacts. AECR-Net and DEA-Net, both CNN-based methods, struggle to capture global context, resulting in detail loss. Compared with DehazeFormer-B and MambaIRv2-S, our model achieves better structural preservation and finer detail restoration, which can be attributed to its improved ability to integrate contextual information.

### 4.4 COMPARISON ON DESMOKEDATA

Fig.4 presents the qualitative desmoking results of various methods on real laparoscopic smoke images. It can be observed that DCP still suffers from severe color distortions. AECR-Net and MambaIRv2-S tend to oversmooth the recovered regions, leading to loss of structural details. Al-

Table 2: Ablation analysis of key modules on the synthetic dataset and the DesmokeData benchmark.

| Methods | Synthetic Dataset | | | DesmokeData | | | Overhead | |
|---|---|---|---|---|---|---|---|---|
| | SSIM↑ | PSNR↑ | CIEDE↓ | SSIM↑ | PSNR↑ | CIEDE↓ | Params↓ | MACs↓ |
| Mamba | 0.967 | 29.423 | 2.631 | 0.876 | 26.408 | 4.425 | 1.55M | 15.72G |
| Vmamba | 0.967 | 29.558 | 2.674 | 0.878 | 26.343 | 4.335 | 2.05M | 20.72G |
| Spatial-Mamba | 0.968 | 29.553 | 2.608 | 0.883 | 26.185 | 4.464 | 1.61M | 16.79G |
| DefMamba | 0.969 | 29.445 | 2.612 | 0.880 | 26.401 | 4.365 | 2.29M | 19.56G |
| FR-FFN→MLPFFN | 0.971 | 29.646 | 2.564 | 0.886 | 26.785 | 4.221 | 1.82M | 19.74G |
| FR-FFN→EDFFN | 0.971 | 29.911 | 2.488 | 0.885 | 26.796 | 4.135 | 1.83M | 18.19G |
| $S(h_t)[x] \rightarrow S(h_t)[h]$ | 0.968 | 29.347 | 2.679 | 0.877 | 26.213 | 4.508 | 1.69M | 18.62G |
| $h_t$ Fusion → $x_t$ Fusion | 0.969 | 29.659 | 2.577 | 0.880 | 26.351 | 4.312 | 1.69M | 18.62G |
| **Ours** | **0.972** | **30.393** | **2.354** | **0.888** | **27.072** | **4.064** | 1.69M | 18.62G |

though DehazeFormer-B and DEA-Net can preserve edge contours to some extent, they struggle to completely eliminate smoke in challenging regions. In contrast, our method consistently produces cleaner outputs while maintaining clear anatomical boundaries, demonstrating stronger performance in challenging real surgical scenarios. The quantitative results in Table. 1 further confirm the superior performance of our approach.

## 4.5 ABLATION STUDY

To evaluate the effectiveness of HG-SSM, we replace it with the SSM modules from Mamba, VMamba, Spatial-Mamba and DefMamba respectively, while keeping the rest of the architecture unchanged. As shown in the gray-highlighted area of Table. 2, our HG-SSM consistently outperforms these alternatives in desmoking performance. This demonstrates its superior capacity for feature modeling, attributed to its flexible state fusion mechanism that promotes more effective contextual information flow.

Next, we evaluate the effectiveness of the FR-FFN module. Replacing FR-FFN with either MLP-FFN or EDFFN (Kong et al., 2025) results in reduced desmoking performance. Although EDFFN also operates in the frequency domain, it adopts a uniform frequency modulation strategy. In contrast, FR-FFN employs multi-band decomposition with fine-grained weighting, which better facilitates the recovery of structural information.

Furthermore, we validate the importance of the input-guided dynamic sampling strategy. When the sampling function $S(h_t|x)$ is replaced with a state-based sampling variant $S(h_t|h)$, the model performance decreases. Additionally, applying adaptive sampling and fusion directly on the original input $x_t$ does not lead to meaningful improvements in feature representation. Fig. 5b shows the hidden states $h_t$ before and after fusion in the final HG-Mamba block. It can be observed that the input-guided dynamic sampling strategy enables the state map to better focus on smoke-affected regions, leading to more effective smoke perception and removal. In contrast, the unguided version fails to reconstruct the spatial structure effectively. These qualitative and quantitative results collectively demonstrate the value of the proposed heuristic-guided state fusion, which allows for flexible state selection and adaptive weighting to enhance smoke awareness and restoration quality.

## 5 CONCLUSION

In this work,, we propose HG-Mamba, a novel backbone for laparoscopic image desmoking, which combines a Heuristic-Guided State Space Model (HG-SSM) and a Frequency Refine Feed-Forward Network (FR-FFN). HG-SSM introduces input-guided dynamic sampling and heuristic-guided state fusion for flexible context modeling, while FR-FFN improves desmoking via multi-frequency decomposition and dynamic modulation. Experiments on synthetic and real datasets show that HG-Mamba consistently outperforms existing methods. Moreover, with only 1.69M parameters and reduced MACs compared to previous models, HG-Mamba demonstrates a lightweight design and low computational cost, indicating strong potential for practical deployment in laparoscopic image desmoking.

## 5.1 REPRODUCIBILITY CHECKLIST

This paper

- Includes a conceptual outline and/or pseudocode description of AI methods introduced (yes/partial/no/NA)**yes**
- Clearly delineates statements that are opinions, hypothesis, and speculation from objective facts and results (yes/no)**yes**
- Provides well marked pedagogical references for less-familiare readers to gain background necessary to replicate the paper (yes/no)**yes**

Does this paper make theoretical contributions? (yes/no)**no**

- All assumptions and restrictions are stated clearly and formally. (yes/partial/no/NA)**NA**
- All novel claims are stated formally (e.g., in theorem statements). (yes/partial/no/NA)**NA**
- Proofs of all novel claims are included. (yes/partial/no/NA)**NA**
- Proof sketches or intuitions are given for complex and/or novel results. (yes/partial/no/NA)**NA**
- Appropriate citations to theoretical tools used are given. (yes/partial/no/NA)**NA**
- All theoretical claims are demonstrated empirically to hold. (yes/partial/no/NA)**NA**
- All experimental code used to eliminate or disprove claims is included. (yes/partial/no/NA)**NA**

Does this paper rely on one or more datasets? (yes/no)**yes**

If yes, please complete the list below.

- A motivation is given for why the experiments are conducted on the selected datasets (yes/partial/no/NA)**yes**
- All novel datasets introduced in this paper are included in a data appendix. (yes/partial/no/NA)**no**
- All novel datasets introduced in this paper will be made publicly available upon publication of the paper with a license that allows free usage for research purposes. (yes/partial/no/NA)**yes**
- All datasets drawn from the existing literature (potentially including authors' own previously published work) are accompanied by appropriate citations. (yes/no/NA)**yes**
- All datasets drawn from the existing literature (potentially including authors' own previously published work) are publicly available. (yes/partial/no/NA)**yes**
- All datasets that are not publicly available are described in detail, with explanation why publicly available alternatives are not scientifically satisficing. (yes/partial/no/NA)**yes**

Does this paper include computational experiments? (yes/no)**yes**

If yes, please complete the list below.

- Any code required for pre-processing data is included in the appendix. (yes/partial/no/NA)**yes**.
- All source code required for conducting and analyzing the experiments is included in a code appendix. (yes/partial/no)**yes**
- All source code required for conducting and analyzing the experiments will be made publicly available upon publication of the paper with a license that allows free usage for research purposes. (yes/partial/no)**yes**
- All source code implementing new methods have comments detailing the implementation, with references to the paper where each step comes from (yes/partial/no)**yes**

- If an algorithm depends on randomness, then the method used for setting seeds is described in a way sufficient to allow replication of results. (yes/partial/no/NA)**partial**

- This paper specifies the computing infrastructure used for running experiments (hardware and software), including GPU/CPU models; amount of memory; operating system; names and versions of relevant software libraries and frameworks. (yes/partial/no)**partial**

- This paper formally describes evaluation metrics used and explains the motivation for choosing these metrics. (yes/partial/no)**yes**

- This paper states the number of algorithm runs used to compute each reported result. (yes/no)**yes**

- Analysis of experiments goes beyond single-dimensional summaries of performance (e.g., average; median) to include measures of variation, confidence, or other distributional information. (yes/no)**no**

- The significance of any improvement or decrease in performance is judged using appropriate statistical tests (e.g., Wilcoxon signed-rank). (yes/no)**no**

- This paper lists all final (hyper-)parameters used for each model/algorithm in the paper's experiments. (yes/no)**yes**

- This paper states the number and range of values tried per (hyper-) parameter during development of the paper, along with the criterion used for selecting the final parameter setting. (yes/partial/no/NA)**NA**

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

# A APPENDIX

## A.1 USING AN LLM TO HELP WITH PAPER WRITING

We used ChatGPT to assist with grammar checking and improving the clarity of writing.

## A.2 LOSS ABLATION

This study employs both the $L_1$ loss and Frequency-domain Contrastive Regularization (FCR) loss to train the smoke removal network. The L1 loss measures the pixel-wise reconstruction error between the predicted desmoked image and the input smoky image. The FCR loss introduces the Discrete Fourier Transform (DFT) as a frequency-domain encoder to construct a contrastive space, effectively capturing the salient frequency characteristics between positive and negative samples. In our setting, the smokeless image $I_{sl}$ is considered the positive sample, while the smoky image $I_{s_i}$ is the negative sample, allowing the contrastive loss to be formulated in the frequency domain as follows:

$$L_{FCR} = \frac{1}{n} \Sigma_{i=1}^{n} \frac{\|\mathcal{F}(I_{sl}) - \mathcal{F}(\hat{I}_{sl})\|_1}{\|\mathcal{F}(I_{s_i}) - \mathcal{F}(\hat{I}_{sl})\|_1}, \quad (10)$$

where $\hat{I}_{sl}$ denotes the output of the model, $n$ is the number of negative samples, and $\mathcal{F}$ represents the Discrete Fourier Transform (DFT).

Table 3 shows the impact of different loss configurations on model performance. The combination of L1 and FCR leads to the best overall performance.

Table 3: An ablation study was conducted on synthetic dataset to investigate the effects of different loss components.

| $L_1$ | $L_{FCR}$ | SSIM↑ | PSNR↑ | CIEDE↓ |
|:---:|:---:|:---:|:---:|:---:|
| ✓ | | 0.969 | 29.610 | 2.597 |
| ✓ | ✓ | **0.972** | **30.393** | **2.354** |

### A.3 Effect of Smoke Removal on Downstream Segmentation

To further evaluate the impact of smoke removal on downstream tasks, we adopt surgical instrument segmentation as a representative case study. The EndoVis2018 dataset, which contains smoke-free laparoscopic images, serves as the evaluation benchmark. To simulate realistic degradation, synthetic smoke is added to the dataset to generate paired smoky/smokeless image samples. We then directly apply our model, as well as competing models pretrained on synthetic smoke data, to perform smoke removal. Finally, the pretrained MedSAM is used to segment surgical instruments from the smoky images, smokeless images, and the de-smoked outputs produced by each model.

Table 4 presents the segmentation performance across different inputs. All methods enhance intra-operative visibility to some extent, alleviating performance degradation caused by smoke occlusion. Compared to existing approaches, our method achieves more competitive results while maintaining lower model complexity and computational cost. For instance, although it slightly outperforms DEA-Net and SGDN, our model is more lightweight in terms of parameters and requires fewer MACs (Ours: 1.69M, 18.62G MACs; DEA-Net: 3.65M, 24.68G MACs; SGDN: 11.09M, 41.16G MACs).

Table 4: Surgical Instrument Segmentation Performance on EndoVis2018 under Different Input Conditions. 'Smoky' indicates images with smoke used as input, and 'Smokeless' indicates the corresponding smoke-free images.

| Methods | DICE↑ | IOU↑ |
|:---:|:---:|:---:|
| Smoky | 0.633 | 0.468 |
| Smokeless | **0.805** | **0.678** |
| DCP | 0.774 | 0.636 |
| Cyclic-DeGAN | 0.738 | 0.589 |
| FFA-Net | 0.782 | 0.645 |
| AECR-Net | 0.768 | 0.627 |
| DS-CycleGAN | 0.701 | 0.545 |
| DehazeFormer-B | 0.791 | 0.661 |
| DEA-Net | 0.793 | 0.662 |
| ConvIR | 0.778 | 0.640 |
| MB-Taylor-B V2 | 0.790 | 0.656 |
| SGDN | 0.792 | 0.660 |
| MambaIRv2-B | 0.785 | 0.650 |
| Ours | 0.794 | 0.662 |

### A.4 Inference Time Evaluation

Table 1 summarizes the parameter count and computational cost (MACs) of HG-Mamba compared with state-of-the-art desmoking models. HG-Mamba achieves a lightweight design with only 1.69M parameters and low computational cost of 18.62G MACs, significantly lower than most existing methods (e.g., MambaIRv2-S: 9.63M parameters, 192.91G MACs).

Inference times are further evaluated on an NVIDIA A100 GPU (see Table 5). HG-Mamba demonstrates competitive inference speed (46 ms for batch size 1 and 37.5 ms for batch size 4), while

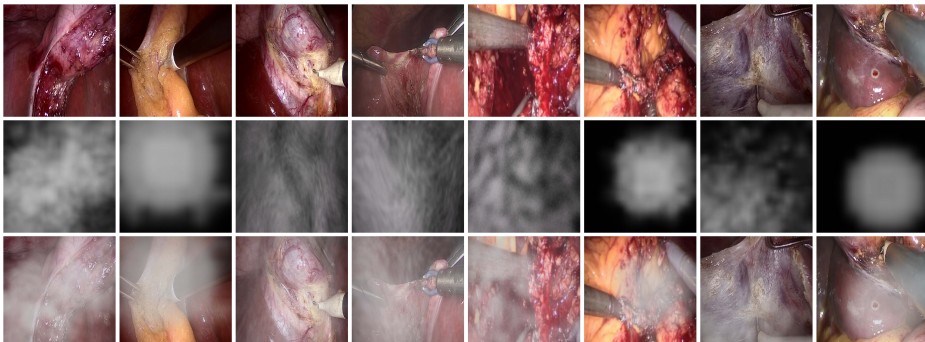

Figure 6: Examples from the synthetic dataset: smokeless images (top), Blender-generated smoke masks (middle), and corresponding synthetic smoky images (bottom).

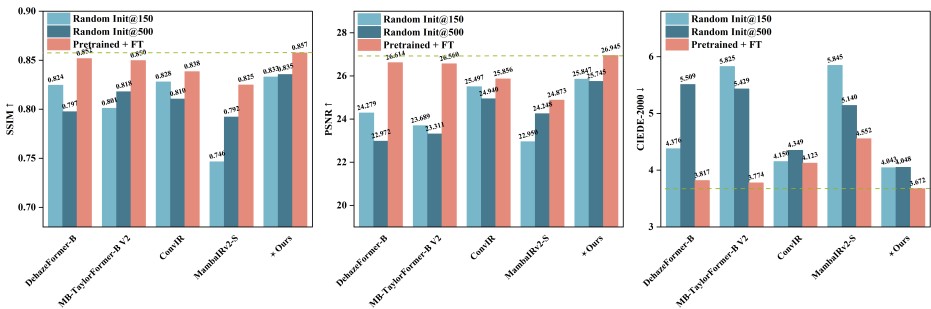

Figure 7: Performance comparison on 30% of the DesmokeData training set: fine-tuning for 50 epochs with synthetic pre-training vs. training from random initialization for 150 and 500 epochs.

maintaining high desmoking quality. This indicates that HG-Mamba achieves a practical trade-off between computational efficiency and restoration performance.

Table 5: Average inference time of desmoking models on the synthetic dataset (input size $224 \times 224$).

| Methods | DehazeFormer | MB-Taylor-B V2 | SGDN | MambaIRv2-S | Ours |
|---|---|---|---|---|---|
| Batch=1 | 23.5 ms | 183.5 ms | 40.0 ms | 420.0 ms | 46.0 ms |
| Batch=4 | 12.0 ms | 65.0 ms | 19.5 ms | 335.0 ms | 37.5 ms |

## A.5 EFFECT OF SYNTHETIC PRE-TRAINING

Fig. 7 shows the performance of models fine-tuned on 30% of the DesmokeData training set using weights pre-trained on the synthetic dataset. For comparison, we report results from models trained from random initialization, including the best checkpoints within ≤150 epochs and those achieving the best performance between 150 and 500 epochs. With such a limited subset, training for more epochs from random initialization often leads to overfitting and does not consistently improve performance. In contrast, synthetic pre-training provides additional diversity and training signals, yielding clear and stable gains under these limited-data conditions. This highlights the value of synthetic datasets, which generate diverse and controllable smoke patterns—where parameters such as density, vorticity, and heat are set in Blender—and serve as an economical and efficient supplement to scarce paired data, helping to alleviate data limitations in laparoscopic smoke removal research. Examples of the synthetic dataset can be seen in Fig. 6.

## A.6 VIDEO DEMONSTRATIONS

Two supplementary videos are provided to visually demonstrate the effectiveness of our smoke removal model. Each video shows the original smoky images on the left and the corresponding de-smoked outputs from our model on the right, allowing direct visual comparison before and after smoke removal.

The videos, named "Smoky-v.s-Desmoked1.mp4" and "Smoky-v.s-Desmoked2.mp4", are based on the DesmokeData dataset, which contains paired smoky and smoke-free images extracted from real surgical videos, reflecting authentic intraoperative scenarios. In the videos, the left side displays the original smoky images, while the right side shows the de-smoked outputs generated by our model. As illustrated, the proposed method effectively removes surgical smoke, restores structural details, and significantly enhances intraoperative visibility.

