# OpenReview forum: "HG-Mamba: Heuristic-Guided State Space Model for Laparoscopic Image Desmoking"
_ICLR.cc/2026/Conference — ICLR 2026 Conference Withdrawn Submission_

### Official Review · Reviewer_nPXX · 2025-10-31

**Soundness:** 2
**Presentation:** 2
**Contribution:** 2
**Rating:** 2
**Confidence:** 4

**Summary:**

This method describes a mamba-based method for Laparoscopic Image Desmoking.They proposed to use deformable vmamba block and frequency-refine FFN. The proposed method shows positive results on synthethic and real datasets.

**Strengths:**

+ The manuscript is well-organized.

**Weaknesses:**

+ The proposed method is not well-motivated. What is the unique challenge of Laparoscopic Image Desmoking comparing with natural image desmoking? Why do we need deformable sampling for such a task?
+ The technical noverlty of this manuscript is limited. It is already a common practice in the low-level community to involve frequency-awareness to the model design([1-5], to name just a few). Deformable sampling is also not new[6]. It is a neat engineering solution to stack the most powerful component to form a powerful model, but they are not suitable for a research paper in top venue.
+ The author should add more comparision with recent desmoke and dehaze methods.
+ The author should demonstate generalization performance. Can the model trained on syntheic data generalize on real-world scenarios? Can the proposed laparoscopic image desmoking model work in other medical image desmoking task?

[1] Revitalizing Convolutional Network for Image Restoration, TPAMI 25

[2] Image Restoration via Frequency Selection, TPAMI 25

[3] High-Resolution Document Shadow Removal via A Large-Scale Real-World Dataset and A Frequency-Aware Shadow Erasing Net, ICCV 23

[4] Frequency and Spatial Dual Guidance for Image Dehazing, ECCV 24

[5] Unveiling Advanced Frequency Disentanglement Paradigm for Low-Light Image Enhancement, ECCV 24

[6] DefMamba: Deformable Visual State Space Model, CVPR 25

**Questions:**

Please refer to the weakness part.

---

### Official Review · Reviewer_qmDz · 2025-10-31

**Soundness:** 2
**Presentation:** 2
**Contribution:** 2
**Rating:** 4
**Confidence:** 5

**Summary:**

This paper proposes HG-Mamba, a Mamba-based image desmoking network for laparoscopic surgery. It comprises a Heuristic-Guided State Space Model (HG-SSM) for adaptive spatial context modeling and a Frequency Refine Feed-Forward Network (FR-FFN) for better feature representation by fusing frequency information. Experiments are conducted on synthetic and real-world data, showing the performance improvement.

**Strengths:**

1. Heuristic-Guided State Space Model (HG-SSM) utilizes dynamic sampling and state fusion for flexible context modeling.
2.  Frequency Refine Feed-Forward Network (FR-FFN) utilizes multi-frequency decomposition and dynamic weighting to improve desmoking performance.
3. A large synthetic smoke dataset is constructed. Experiments demonstrate that the proposed method gains superior performance over state-of-the-art ones.

**Weaknesses:**

1. The motivation and reasons for the effectiveness of HG-SSM still require further discussion. It's somewhat like a combination of ideas from DefMamba and deformable convolution.
2. The proposed method needs to be validated on more similar tasks (such as deraining and dehazing).
3. The inference time can be given in Tables 1 and 2.
4. The improvement in visual effects in Figures 3 and 4 does not appear to be significant.
5. From Table 3 in Suppl., Frequency-domain Contrastive Regularization (FCR) loss plays a crucial role in improving performance. How do the compared methods perform when trained using this loss?
6. Lack related citations. e.g., [1,2]. It would be better to validate the proposed method on SurgClean [2] dataset.

[1] Self-Supervised Video Desmoking for Laparoscopic Surgery.
[2] Benchmarking Laparoscopic Surgical Image Restoration and Beyond.

**Questions:**

Please see Weaknesses.

---

### Official Review · Reviewer_TH4y · 2025-11-03

**Soundness:** 2
**Presentation:** 2
**Contribution:** 2
**Rating:** 2
**Confidence:** 4

**Summary:**

The authors propose Heuristic-Guided Mamba (HG-Mamba), which extends Mamba by integrating spatial and frequency domain improvements. HG-Mamba comprises two key components: a Heuristic-Guided State Space Model (HG-SSM), which performs input-guided dynamic sampling and flexible state fusion to enable adaptive spatial context modeling; and a Frequency Refine Feed-Forward Network (FR-FFN), which conducts multi-band frequency decomposition and adaptive weighting to enhance frequency-domain representations.

**Strengths:**

According to the authors, the paper's contributions are summarized in the following:

1. HG-SSM is proposed, utilizing dynamic sampling and state fusion.
2. FR-FFN is designed to enhance feature representation.
3. HG-Mamba is a novel laparoscopic de-smoking backbone combining HG-SSM and FR-FFN.
4. The authors proposed a large synthetic smoke dataset.

Overall, the comparison results do not show much significant improvement; however, the overhead results look very promising.

**Weaknesses:**

1. Regarding the 4th stated contribution, it is unclear whether the authors are claiming to be the first to propose the smoke dataset or if the dataset is adapted or derived from previously published work. The authors should explicitly clarify the origin of the dataset, including whether it is newly created or based on an existing dataset that previously released. If it is new, details such as the data collection process, annotation methodology, and dataset availability should be provided to substantiate its novelty and reproducibility.

2. The paper currently lacks any statistical or significance testing to demonstrate whether the reported performance improvements are meaningful compared to existing methods. Without such analyses, it is difficult to assess the robustness and generalizability of the proposed approach. The authors are strongly encouraged to include quantitative significance tests to validate that the observed differences are statistically significant and not due to random variation when comparing with competing methods.

3. The manuscript would be strengthened by a discussion of the limitations of the proposed method. For example, under what conditions does the approach fail or underperform? Are there constraints related to dataset diversity or generalizability to unseen scenarios? Clearly outlining these limitations would not only enhance the scientific rigor of the paper but also provide valuable insights for the proposed work.

**Questions:**

Please refer to the weakness section.

**Details Of Ethics Concerns:**

No Institutional Review Board (IRB) or ethical approval statement is provided, even though the authors mention constructing a dataset as part of their contribution. It is unclear whether this dataset is entirely synthetic or includes real data. The authors should clarify the nature of the dataset and, if real data were used, provide an appropriate IRB or ethics approval statement to ensure compliance with research ethics standards.

---

### Official Review · Reviewer_WKgY · 2025-11-12

**Soundness:** 3
**Presentation:** 3
**Contribution:** 2
**Rating:** 4
**Confidence:** 4

**Summary:**

This paper addresses the removal of laparoscopic surgical smoke to enhance visibility. The authors claimed that existing Mamba-based models are limited by sequential state transitions that restrict spatial correlation modeling in 2D images, and their MLP-based feed-forward networks do not consider frequency-domain characteristics needed for smoke removal. Therefore, they propose HG-Mamba: HG-SSM for input-guided dynamic sampling and fusion of spatially related states (breaking sequential dependencies), and FR-FFN for multi-band frequency decomposition with adaptive weighting.

**Strengths:**

* Parameter efficiency makes the method potentially deployable in settings with limited resources.
* The dual-domain approach (spatial and frequency) is interesting.
* Ablation studies show the contribution of each component

**Weaknesses:**

- Lack of theoratical justification
- Empirical frequency filtering mechanisms.
- Lack of empirical validation of the computational efficacy for the overhead of dynamic sampling
- The validation with synthetic data lowers the authenticity of the experiments

**Questions:**

Have you considered dynamic frequency band selection where the frequency boundaries adapt based on input characteristics rather than using fixed octave-based divisions?

---

### Note · Authors · 2025-11-17

I have read and agree with the venue's withdrawal policy on behalf of myself and my co-authors.